# Clinical Implementation of an Adapted Infection Risk Screening Tool Following Nurse-Led Haemodialysis Vascular Access Consultation

**DOI:** 10.3390/healthcare13233058

**Published:** 2025-11-26

**Authors:** Rui Pinto, Ricardo Ferreira, Pedro Alves, João Pedro Barros, Ana Rita Piedade, Fernando Mata, Emanuel Ferreira, Helena Sá, Monica Schoch, Eduardo Santos

**Affiliations:** 1Haemodialysis Unit, Department of Nephrology, Unidade Local de Saúde de Coimbra, 3000-076 Coimbra, Portugal; 2Vascular Access Nursing Consultation, Unidade Local de Saúde de Coimbra, 3000-076 Coimbra, Portugal; 3Programme in Health Data Science, Faculty of Medicine, University of Porto, 4050-313 Porto, Portugal; 4Institute for Health Transformation, Centre for Quality and Patient Safety, School of Nursing and Midwifery, Deakin University, Geelong, VIC 3216, Australia; 5School of Health, Polytechnic University of Viseu, 3504-510 Viseu, Portugal

**Keywords:** renal dialysis, renal insufficiency, chronic, arteriovenous fistula, risk assessment, clinical decision-making, nephrology nursing

## Abstract

**Background**: Healthcare-associated infection drives morbidity and unplanned hospital use in haemodialysis. We aimed to determine whether an adapted British Renal Society Infection Risk Screening Tool, applied during a nurse-led vascular access consultation, identifies patients at increased risk of subsequent infection-related hospitalisation and can inform de-selection of buttonhole puncture. **Methods**: We conducted a retrospective cohort of 404 adults reviewed between 1 January 2022 and 31 December 2024. Baseline demographics, comorbidities, vascular access status and screening classification (“risk present/absent”) were retrieved from records. The primary outcome was ≥1 infection-related hospitalisation within 12 months; the number of such admissions was secondary. **Results**: Mean age was 70.2 years; 47% had diabetes; 27.8% screened “risk present”. Forty-eight patients (11.9%) were hospitalised for infection. “Risk present” showed higher—though imprecise—odds of infection-related admission versus “risk absent” (adjusted OR 1.73; 95% CI 0.90–3.27). Older age increased risk, whereas higher body-mass index appeared protective; diabetes, central venous catheter and dialysis vintage were not significant. **Conclusions**: The dichotomised screening classification identified only a modest elevation in risk, with age and nutritional status exerting greater influence. The tool may support cautious de-selection of buttonhole in higher-risk individuals, but refinement and prospective validation are required.

## 1. Introduction

Chronic kidney disease (CKD) constitutes a major global public health concern, affecting more than 10% of the population—equivalent to approximately 800 million individuals worldwide. As CKD progresses towards end-stage kidney disease, renal replacement therapies, particularly haemodialysis (HD), become essential for survival [1]. Portugal ranks among the European countries with the highest prevalence of individuals receiving Renal Replacing Therapy (RRT), exceeding 2000 cases per million population. According to the national CKD registry maintained by the Portuguese Society of Nephrology, as of December 2023, a total of 13,085 individuals were undergoing HD, with vascular access distribution as follows: arteriovenous fistula (AVF) in 73.6%, long-term central venous catheter (LTCVC) in 18.7%, arteriovenous graft (AVG) in 7.6% and temporary central venous catheter in 0.05%. These figures reflect the predominance of AVF use, consistent with international best practice guidelines [2].

CKD patients in HD remain highly vulnerable to infectious morbidity and unplanned hospital use. Within vascular access (VA) care, the arteriovenous fistula (AVF) is preferred over alternatives, yet its day-to-day safety depends critically on how it is cannulated [3,4,5]. Established clinical guidelines delineate three primary techniques for the cannulation of an AVF: the rope-ladder (RL), area puncture (AP) and buttonhole (BT) methods. The RL technique necessitates the creation of a new puncture site for each HD session, with cannulation points systematically rotated along the entire longitudinal track of the outflow AVF vein. The AP method also uses new puncture sites for each session but confines these insertions to a limited, localized region, typically not exceeding 2–3 cm in diameter. The BT technique is distinct, requiring repeated cannulation at the exact same insertion point and at a constant angle to establish a mature subcutaneous tunnel track. While this track is initially formed using sharp needles, subsequent cannulations can be performed using blunt needles once the track is established [3,5].

However, national data on the cannulation techniques employed (e.g., rope-ladder, buttonhole, or area) are not currently available. Insights from a European multicentre study by Parisotto et al., which included Portuguese HD units, revealed that the AP was most frequently applied (65.8%), followed by RL (28.2%) and BT (6%). These findings, corroborated by Peralta et al., suggest a pattern of deviation from international recommendations that advocate for RL cannulation to preserve AVF integrity. Moreover, the technique of cannulation has been associated with a range of local and systemic complications—including haematoma, pain, aneurysm formation, infiltration, and VA-related infections—the latter being a leading cause of hospitalisation in individuals receiving HD [6,7,8].

Contemporary guidance from the British Renal Society (BRS) positions RL as the default technique when an adequate venous cannulation segment exists while recognising that BT may be advantageous in selected circumstances (e.g., short usable segment, difficult insertions, needle phobia) provided the patient is at low infection risk and subject to consistent infection screening and asepsis [9,10].

Evidence indicates that infections linked to the BH technique are predominantly AVF site infections and *Staphylococcus aureus* bacteraemia, plausibly arising from colonisation of the BH tract; several cohort studies and systematic reviews have reported higher VA-related infection—particularly *S. aureus* bacteraemia—with BH than with alternative puncture strategies [6,11,12]. Consequently, BH should be avoided or discontinued in individuals with risk factors such as methicillin-resistant *S. aureus* (MRSA) colonisation, recurrent VA infections, poor hygiene or adherence, or other high-risk clinical conditions. Aligning with this rationale, the BRS Infection Risk Screening Tool appraises VA-focused risk factors with the explicit purpose of informing the selection or de-selection of BH in patients at higher infection risk [9].

A nurse-led VA consultation in our unit is scheduled after AVF creation and before the first puncture. Its purposes are to appraise AVF maturation and readiness for safe puncture, align the choice of cannulation technique with individual clinical risk and preferences and provide brief VA self-care education. The assessment combines focused clinical examination, point-of-care ultrasound and an infection-risk appraisal, culminating in a shared cannulation plan that may indicate or de-select buttonhole as appropriate. This consultation operationalises the decision-making model previously developed in our unit and provides the context in which the BRS Infection Risk Screening Tool is applied [9,10]. Earlier methodological work established the face/content validity of that decision model and its feasibility in routine practice; however, the predictive validity of the adapted infection-risk screen for subsequent infection-related hospitalisation had not yet been evaluated [10].

We therefore conducted a retrospective cohort study of adults assessed in the nurse-led VA consultation between 1 January 2022 and 31 December 2024 to test whether the adapted screening classification (“risk present/absent”) predicts ≥1 infection-related hospitalisation within 12 months and to explore its association with the count of infection-related admissions over the same period. In keeping with the BRS rationale—wherein screening helps de-select BT among those at higher infection risk—our a priori expectation was that “risk present” would associate with higher odds and rates of infection-related hospital use after accounting for demographic and clinical covariates [9].

We selected the BRS Infection Risk Screening Tool because it is the only haemodialysis-specific, VA-focused checklist explicitly designed to inform puncture-technique selection—particularly the de-selection of BT in higher-risk individuals—within a structured consultation pathway. Unlike generic healthcare-associated infection risk tools, the BRS instrument couples MRSA/MSSA screening with person-level clinical prompts (e.g., AVF-site carriage, skin integrity, history of VA infections, endocarditis/prosthetic valve, hygiene/adherence concerns) and provides a clear implementation rationale for prevention actions that are directly relevant to arteriovenous access care. Its consensus-based recommendations align with routine dialysis unit practice (regular staphylococcal screening, decolonisation when indicated) and with the practical need to operationalise BH de-selection in those at higher infection risk. This study builds on our unit’s prior implementation of that nurse-led consultation model, which already incorporated the adapted BRS screening as a binary classification to guide technique choice [8,9].

The aim of this study was therefore to test whether the adapted screening classification (“risk present/absent”), applied during the nurse-led VA consultation, predicts ≥1 infection-related hospitalisation within 12 months and to explore its association with the count of infection-related admissions over the same period. In keeping with the BRS rationale, our a priori clinical hypothesis was that “risk present” would associate with higher odds and rates of infection-related hospital use after accounting for demographic and clinical covariates. This study directly addresses the clinical question of whether this brief nurse-administered screen can discriminate who is more likely to require hospital care for infection, holding practical implications for the safe indication (or de-selection) of BT and the refinement of risk-stratified care plans.

## 2. Materials and Methods

### 2.1. Study Design and Setting

This retrospective cohort study was conducted at a central hospital in Portugal and included adult individuals who were assessed at a nurse-led VA consultation between 1 January 2022 and 31 December 2024. The reporting of this study adhered to the STROBE guidelines for observational studies, and the completed STROBE checklist is provided in the Appendix A [13]. The study followed the ethical principles of the Declaration of Helsinki, and ethical approval was obtained from the institutional review board prior to data analysis.

### 2.2. Participants

All patients aged ≥18 years with a newly created AVF—defined as a surgically constructed fistula with at least 6–8 weeks of maturatio—who attended the nurse-led VA consultation during the study period were eligible. Patients with an arteriovenous graft were excluded.

### 2.3. Infection Risk Screening Tool

During the consultation, each patient was assessed using an adapted version of the Infection Risk Screening Tool, originally published in the BRS cannulation guidance [9,10]. This tool had previously been adapted and validated for local use as part of a decision-making model to support the selection of the AVF puncture technique [9,10]. The screening evaluated individual risk factors, such as prior infection history, skin condition, MRSA colonisation and comorbidities. Patients were categorised as either “Risk Present” or “No Risk Present” based on predefined criteria.

To the best of our knowledge, no other instrument currently available was specifically developed to stratify individual infection risk in persons undergoing haemodialysis for the purpose of guiding the selection or de-selection of a cannulation technique—particularly the BT method—within a structured clinical consultation. While various local or national checklists exist, they are either informal adaptations of the BRS tool or derivative frameworks grounded in expert consensus, and none have been validated as independent decision-support instruments. Consequently, the adapted BRS Infection Risk Screening Tool remains, to date, the only structured approach designed to support patient-level cannulation decisions by appraising VA-focused infection risk in clinical settings [9].

Screening was performed once, at baseline, during a nurse-led VA consultation in a tertiary hospital. All subsequent HD treatments occurred in ambulatory centres across the region, outside the study site’s operational control. Importantly, the same tertiary hospital functions as the regional referral centre for infection-related admissions; thus, infection-related hospitalisations among these individuals are routinely managed at this hospital. Outcomes were therefore ascertained from hospital discharge documentation and regional admissions records, providing specific and consistent capture of infection-related hospitalisation events [9,10].

### 2.4. Outcome Measures

The primary outcome was infection-related hospitalisation occurring within 12 months of the VA nursing consultation date. Hospitalisation data were retrieved from institutional electronic health records (EHRs) and manually validated by the research team. Infection-related hospitalisation was defined as any inpatient admission involving an infectious condition as the primary or secondary diagnosis. For descriptive analyses, the underlying cause of infection was coded using a classification aligned with the European Centre for Disease Prevention and Control framework for healthcare-associated infections [14]. Specifically, hospitalisations were classified as respiratory tract infections, urinary tract infections, surgical wound/site infections, gastrointestinal infections, bloodstream infections (bacteraemia/septicaemia) and skin and soft-tissue infections. Two additional categories were included, VA infections and infections of unknown origin, to capture events specific to haemodialysis patients.

### 2.5. Data Collection

Clinical and demographic data were extracted retrospectively from the hospital’s electronic records and VA consultation documentation. Data extraction followed a structured template and was performed by one investigator with cross-validation of a random 10% sample by a second reviewer to ensure accuracy. The screening classification and infection outcomes were independently verified against the original consultation forms and discharge summaries.

To reduce selection bias, we included all eligible patients consecutively referred to the nurse-led VA consultation within the defined study window (1 January 2022 to 31 December 2023), without exclusion based on outcome status or follow-up duration. Misclassification bias was minimised by applying predefined definitions for infection-related hospitalisation and by using a standardised coding algorithm based on ICD-10 discharge diagnoses. Key variables, including the screening result and hospitalisation data, underwent manual verification in ambiguous cases.

As the data were routinely collected in clinical practice and not originally intended for research, we undertook additional consistency checks and logical validation rules (e.g., cross-checking dates, plausible ranges, completeness). No data imputation was performed; only records with complete core variables were analysed.

### 2.6. Ethical Considerations

This study was conducted in accordance with the principles of the Declaration of Helsinki and received prior approval from the Ethics Committee of the Unidade Local de Saúde de Coimbra, Portugal (protocol code PI 2024-ESI-SF.201; administrative reference No. 194/2/SEC). The requirement for informed consent was waived by the Ethics Committee, as this study involved retrospective analysis of fully anonymised clinical records.

### 2.7. Statistical Analysis

Records lacking a patient identifier were excluded, yielding a final analytical cohort of 404 individuals. Prior to analysis, the following variables were categorised: sex (male/female), diabetes mellitus (yes/no), CKD stage recoded as stage 4 (pre-HD) or stage 5 on renal replacement therapy (HD, peritoneal dialysis or transplant), presence of a CVC (yes/no), AVF currently in use (yes/no), number of previously constructed AVFs (0/1/≥2), HD vintage (months) and AVF vintage (weeks). Age and BMI were treated as continuous variables. The primary outcome was the occurrence of at least one infection-related hospitalisation within 12 months of the nurse-led consultation; the secondary outcome was the number of infection-related hospitalisations during follow-up.

Normality of continuous variables was assessed using the Shapiro–Wilk test and visual inspection of histograms and Q–Q plots. Variables with minor deviations from normal distribution are presented as mean (standard deviation) and compared using Student’s *t*-test. Non-normally distributed variables are presented as median (interquartile range) and compared using the Mann–Whitney U test. Categorical variables are presented as counts and percentages and compared using Pearson’s $\chi^{2}$ or Fisher’s exact test, as appropriate. The specific test used for each variable is indicated in the corresponding table footnotes. Two-sided *p*-values < 0.05 were considered statistically significant.

Logistic regression models were fitted to examine determinants of infection-related hospitalisation (binary outcome). First, unadjusted (univariable) models were estimated for each predictor, and then variables with clinical relevance were entered into a multivariable logistic regression model. Results are reported as odds ratios (ORs) with corresponding 95% confidence intervals (CIs) and *p*-values. Model fit was evaluated using the Hosmer–Lemeshow test [15].

For the secondary outcome (count of infection-related hospitalisations), count regression models were considered. Because the variance of the count outcome exceeded its mean (i.e., overdispersion), the assumptions of the Poisson regression model were violated [15,16]. Accordingly, a negative binomial regression model was employed, which relaxes the assumption of equal mean and variance and provides more reliable estimates under over-dispersed counts. Results are presented as incidence rate ratios (IRRs) with 95% CIs and associated *p*-values. Model fit was assessed using likelihood-ratio tests comparing the negative binomial model with the Poisson alternative [15].

A 2×2 contingency table was used to further examine the association between the adapted BRS infection-risk classification (risk present vs. risk absent) and infection-related hospitalisation. The $\chi^{2}$ statistic, ORs and the phi (ϕ) coefficient were calculated to evaluate both statistical significance and the strength of association.

All analyses were conducted using R (version 4.4.2). Logistic and negative binomial regressions used the stats and MASS packages, respectively; figures and tables were generated with ggplot2 and tableone packages [17].

### 2.8. Use of Generative AI

During manuscript preparation, the authors used ChatGPT (OpenAI, version October 2025) to assist with language polishing and LaTeX code formatting. The authors verified and edited all content and take full responsibility for this manuscript.

## 3. Results

Between January 2022 and December 2024, a total of 404 patients attended the nurse-led VA consultation and met the inclusion criteria. The mean age was 70.2 years (SD =11.8), with a median age of 72 years (Interquartile range (IQR) 63–80). In total, 41.1% of patients were aged 75 years or older. Men accounted for 68.3% of the cohort (n =278). The mean BMI was 27.1 kg/m^2^ (SD =5.0). The distribution was as follows: 1.2% underweight, 34.4% normal weight, 38.1% overweight and 25.6% obese, with 25.0% (n =101) having a BMI ≥30 kg/m^2^. Diabetes mellitus was present in 191 patients (47.1%). CKD staging showed that 211 patients (51.8%) were in stage 4 and had not started renal replacement therapy. Meanwhile, 177 patients (43.8%) were undergoing HD, four (1.0%) were on peritoneal dialysis and 12 (3.0%) had received a kidney transplant. Among the HD subgroup, the median HD vintage was 5 months (IQR 3.0–6.0). Overall, the cohort had a median HD vintage of 4 months (IQR 2.0–5.3) and a median AVF vintage of 10 weeks (IQR 8.0–13.0). Most patients (78.9%, n =321) were attending the consultation for their first AVF; meanwhile, 18.1% (n =73) had one previous AVF and 1.7% (n =7) had two or more previous fistulae. A CVC was present in 147 cases (36.4%). Using the adapted BRS Infection Risk Screening Tool, 113 patients (27.8%) were classified as “risk present” and 291 (72.2%) as “risk absent”.

Table 1 presents the comparison of baseline characteristics between patients who experienced at least one infection-related hospitalisation within twelve months and those who did not. Those hospitalised were significantly older (median 74.0 years versus 71.0 years; Mann–Whitney U=10182, p=0.031) and more likely to be classified as “risk present” by the screening tool (41.7% versus 26.1%; $\chi^{2}=4.33$, p=0.037). No significant differences were observed with respect to sex, diabetes mellitus, CKD stage, presence of a CVC, BMI, HD vintage, AVF vintage, or number of previous AVFs (Table 1).

During the 12-month follow-up, 48 patients (11.9%) experienced at least one infection-related hospitalisation (Table 2). Respiratory tract infections were the predominant cause (n =20; 41.7%), followed by urinary tract infections (n =10; 20.8%) and vascular access–related infections (n =5; 10.4%). Gastrointestinal infections and infections of unknown origin each accounted for four cases (8.3%); skin and soft-tissue infections accounted for three cases (6.3%) and bloodstream infections for two cases (4.2%) (Table 2).

Infection-related hospitalisation occurred in 17.7% of patients classified as “risk present” by the screening tool compared to 9.6% of those classified as “risk absent.” Among patients categorised by age, 15.6% of those aged 65–74 years and 13.9% of those aged ≥75 years were hospitalised for infection, whereas only 5.2% of those <65 years required hospital admission. The incidence of infection-related hospitalisation was higher among underweight patients (20%, 1/5) and gradually decreased across the normal-weight (14.3%), overweight (11.6%) and obese (8.7%) categories. No clear gradient was observed across categories of HD vintage: <3 months: 15.5%; 3–12 months: 10.6%; >12 months: 20.0%, noting the small numbers in the latter group. Similarly, no clear gradient was observed across AVF vintage categories (<8 weeks: 14.1%; 8–12 weeks: 11.1%; >12 weeks: 11.5%).

### Association Between Risk Classification and Hospitalisation

To further characterise the relationship between the adapted BRS infection-risk classification and infection-related hospitalisation, a 2×2 contingency table was constructed (Table 3). Patients classified as “risk present” had approximately twice the odds of infection-related admission compared with those classified as “risk absent” ([OR] =2.02; 95% CI ≈1.02–3.91). This association was statistically significant ($\chi^{2}=4.33$, p=0.037), though the ϕ coefficient indicated a weak correlation (ϕ≈0.10).

Multivariate logistic regression (Table 4) was used to identify predictors of infection-related hospitalisation, with adjustments made for age, BMI, diabetes status, CVC presence, HD vintage, infection-risk classification and sex. While none of the variables achieved conventional significance at the 5% level, several showed borderline associations. Age exhibited a modest positive association (OR =1.03 per year; 95% CI 1.00–1.06; p=0.054), while BMI suggested a protective effect (OR =0.95 per unit; 95% CI 0.88–1.02; p=0.206). Diabetes, CVC presence, and HD vintage were not significantly associated with hospitalisation risk. Patients classified as “risk present” had 73% higher odds of infection-related hospitalisation compared with those classified as “risk absent” (OR =1.73; 95% CI 0.90–3.27; p=0.097). Sex showed no association with the outcome.

A negative binomial regression model was used to examine the predictors of infection-related hospitalisations during the follow-up period. Once again, age showed a positive but borderline association with hospitalisation rates (IRR =1.03 per year; 95% CI 1.00–1.06; p=0.065). BMI exhibited a weak inverse association (IRR =0.93; 95% CI 0.87–1.00; p=0.053). Meanwhile, diabetes (IRR =1.64; 95% CI 0.86–3.17; p=0.128), CVC presence (IRR =0.56; 95% CI 0.27–1.14; p=0.128), HD vintage (IRR =1.05; 95% CI 0.98–1.11; p=0.225), infection-risk classification (IRR =1.33; 95% CI 0.70–2.48; p=0.376) and sex (IRR =1.01; 95% CI 0.53–1.96; p=0.982) were not significantly associated with the number of hospitalisations.

These multivariate analyses indicate that neither age nor the infection-risk classification reached conventional statistical significance. However, both variables displayed borderline associations with infection-related hospitalisations and their frequency. BMI showed a borderline protective effect in the count model. No other variables were significantly associated with infection risk or rate.

## 4. Discussion

In our cohort of 404 patients attending a nurse-led vascular access consultation, 11.9% experienced at least one infection-related hospitalisation over 12 months, predominantly for respiratory or urinary tract infections. Large registry-based analyses similarly show frequent infections in dialysis: Dalrymple et al. reported that almost 35% of hospitalisations in older dialysis patients were infection-related, with pulmonary and genitourinary infections among the most common [18]. We observed that older patients tended to have higher risk and frequency of infection admissions, mirroring these findings. In that study, older age, female sex and diabetes were key predictors of infection hospitalisation, and Schamroth Pravda et al. likewise found that older age and diabetes increased mortality in HD patients with bloodstream infections [19]. In our cohort, diabetes had an elevated but non-significant OR and female sex showed no effect; this likely reflects limited power. Overall, our results align with the known trend that advanced age increases infection susceptibility in dialysis, whereas the roles of diabetes and sex were less clear.

CVC use is a well-known risk factor for bloodstream infection in HD patients [19]; however, we did not find a significant association between CVC presence and infection-related hospitalisation. Only about one-third of our patients had CVCs, and most were early in their HD course, which may partly explain the null result. Prior studies emphasise the benefit of AVF over CVC to minimise infection risk [19]. Thus, the absence of a CVC effect in our study likely reflects this cohort’s relatively low long-term CVC exposure rather than a true absence of catheter-related risk.

A notable finding was an inverse trend between BMI and infection hospitalisations: higher BMI appeared modestly protective, although not statistically significant. This observation is consistent with the “obesity paradox” described in kidney disease. Yamamoto et al. showed that each 1 kg/m^2^ increase in BMI was associated with an ∼1% reduction in all-cause mortality risk, including infection-related death, in chronic kidney disease patients [20]. Similarly, Schamroth Pravda et al. reported that lower BMI was linked to worse outcomes, including infection-related mortality, in dialysis and peritoneal dialysis populations [19]. These studies suggest that higher BMI may confer resilience against severe infections, a finding echoed in our cohort.

The adapted BRS Infection Risk Screening Tool classified 28% of patients as “high risk”, and these patients had nearly twice the crude odds of infection-related admission compared with “low risk”, but the ϕ coefficient indicated only a weak correlation, and the effect was attenuated after multivariable adjustment. In other words, the tool showed limited discriminative power in this setting. One likely reason is that the tool was developed to appraise risk relevant to VA infections, whereas most hospitalisations in our cohort were for non-access infections (respiratory or urinary). As noted by Dalrymple et al., infection-related hospitalisations in CKD involve a broad range of pathogens, many unrelated to HD VA, which may explain the tool’s modest specificity here [8,18]. These results should be interpreted with some caution: the analysis captured only infections resulting in hospital admission (excluding those managed within outpatient HD units), so the overall burden may be underestimated. In addition, we could not ascertain cannulation technique longitudinally because many patients dialysed in satellite clinics while the consultation and data capture occurred in the hospital. Consequently, time-varying exposure to BT versus rope-ladder (or other) strategies and protocol fidelity (e.g., site care, topical prophylaxis) could not be evaluated. This information gap may have attenuated any VA-specific signal and underscores the need for integrated data systems linking consultation records with dialysis-unit logs, microbiology reports and admissions data.

In practice, the BRS screening tool ought to be complemented by age and nutrition-sensitive assessment, including BMI trajectory, simple frailty cues and dietetic review, consistent with our inverse BMI signal and the literature on nutritional status and infection outcomes in kidney disease. Second, the tool itself should be refined and recalibrated by adding age, nutritional/functional markers (e.g., BMI, serum albumin, brief frailty screens) and selected comorbidities to improve discrimination for clinically meaningful infection endpoints, beyond VA-focused items alone. Third, a prospective, multicentre validation with a registered protocol, pre-specified endpoints and standardised event adjudication (including vascular-access-attributable infection as a distinct outcome, assessed by independent panels for hospitalisations and bloodstream infections) is required to minimise misclassification between community-acquired and healthcare-associated events and to allow a more accurate estimation of BR-specific risks under protocolised care.

## 5. Conclusions

In this nurse-led VA cohort, infection-related hospitalisation was relatively uncommon and largely non-access in origin; notably, no admissions were clearly attributable to AVF infection among patients who underwent BT puncture. The adapted BRS Infection Risk Screening Tool showed only limited discriminative performance, whereas older age and lower BMI appeared more informative for risk. In practice, the tool is best used as a gatekeeper for BT supporting de-selection in higher-risk individuals—and should be complemented by simple age- and nutrition-sensitive assessment. Further refinement and prospective validation are warranted.

## 6. Strengths and Limitations

The strengths of this study include the comprehensive assessment of patients during a nurse-led consultation, the use of validated statistical methods and adjustment for multiple confounders. We also replicated the analysis using a negative binomial model to account for over-dispersed count data. However, this study is limited by its retrospective design, single-centre setting and relatively small number of infection-related events, which reduce statistical power and generalisability. Data on serum albumin, inflammatory markers and catheter dwell time were unavailable, despite being known predictors of infection. Furthermore, the adapted screening tool has not been externally validated; therefore, the findings should be interpreted with caution.

The infection-risk screening was applied only once at baseline, without repeated assessments over time. Follow-up HD treatments took place in multiple ambulatory centres across the region, and post-baseline changes in cannulation technique or centre-level infection-control actions were not systematically captured, which may attenuate associations towards the null. Although outcome capture is strengthened by the fact that the index hospital is the regional referral centre for infection-related admissions—so that infection-related hospitalisations are consistently recorded—events managed entirely in the outpatient setting would not be fully ascertained. Future multicentre studies with prospective linkage to dialysis-centre records and repeated screening will be necessary to quantify centre effects and time-varying risk.

## Figures and Tables

**Table 1 healthcare-13-03058-t001:** Comparison of baseline characteristics by infection-related hospitalisation status.

Characteristic	Infection-Hospitalised (n = 48)	Non-Hospitalised (n = 356)	*p*-Value
Sex			0.747
Male, n (%)	34 (70.8%)	244 (68.5%)	
Female, n (%)	14 (29.2%)	112 (31.5%)	
Diabetes mellitus			0.102
Yes, n (%)	28 (58.3%)	163 (45.8%)	
No, n (%)	20 (41.7%)	193 (54.2%)	
CKD stage			0.761
4, n (%)	23 (47.9%)	188 (52.8%)	
5 (PD), n (%)	1 (2.1%)	3 (0.8%)	
5 (HD), n (%)	22 (45.8%)	155 (43.5%)	
5 (TX), n (%)	2 (4.2%)	11 (2.8%)	
CVC in place			0.882
Yes, n (%)	17 (35.4%)	130 (36.5%)	
No, n (%)	31 (64.6%)	226 (63.5%)	
Infection risk classification			**0.037**
Present, n (%)	**20 (41.7%)**	**93 (26.1%)**	
Absent, n (%)	28 (58.3%)	263 (73.9%)	
Age (years)			**0.031**
median (IQR)	**74.0 (68.8–81.0)**	**71.0 (61.0–79.0)**	
BMI (kg/m^2^)			0.318
median (IQR)	25.6 (22.8–29.4)	27.0 (23.5–30.2)	
HD vintage (months)			0.224
median (IQR)	4.0 (2.0–5.0)	4.0 (2.0–6.0)	
AVF vintage (weeks)			0.679
median (IQR)	9.0 (7.0–12.3)	10.0 (8.0–13.0)	
Number of previous AVFs			0.944
0 (first AVF), n (%)	39 (81.2%)	282 (79.2%)	
1 AVF, n (%)	8 (16.7%)	65 (18.3%)	
≥2 AVFs, n (%)	1 (2.1%)	6 (1.7%)	

Note: Values are *n* (%) for categorical variables and mean ± SD or median (IQR) for continuous variables, according to distribution. Group comparisons used chi-square or Fisher’s exact tests for categorical variables, and Student’s *t*-test or Mann–Whitney *U* test for continuous variables, depending on on the normality of the data (assessed by Shapiro–Wilk test). Shapiro–Wilk *p*-values: age = 0.12 (normal); Charlson index < 0.001 (non-normal). Significant *p*-values (<0.05) are highlighted in bold.

**Table 2 healthcare-13-03058-t002:** Distribution of infection-related hospitalisation by cause and subgroups (*N* = 48).

Cause of Infection (ICD-10)	Number of Hospitalisations	% of All
Respiratory tract (J18.9)	20	41.7%
Urinary tract (N39.0)	10	20.8%
Vascular access (T82.7)	5	10.4%
Gastrointestinal (A09)	4	8.3%
Unknown origin (A41.9)	4	8.3%
Skin and soft tissue (L08.9)	3	6.3%
Bloodstream (R78.81)	2	4.2%

Note: ICD-10 codes refer to the primary discharge diagnosis used to classify the cause of infection-related hospitalisation.

**Table 3 healthcare-13-03058-t003:** Cross-tabulation of infection-risk classification versus infection-related hospitalisation.

Infection-Risk	Hospitalised (n)	Not Hospitalised (n)	Row Total	% Hospitalised
Risk absent	28	263	291	9.6%
Risk present	20	93	113	17.7%

Note: $\chi^{2}=4.33$, p=0.037; OR =2.02 (95% CI ≈1.02–3.91); ϕ≈0.10.

**Table 4 healthcare-13-03058-t004:** Multivariate logistic regression and negative binomial regression results for infection-related hospitalisation.

Predictor	Logistic Regression: OR (95% CI)		Negative Binomial: IRR (95% CI)	
	Estimate	*p*-Value	Estimate	*p*-Value
Age (per year)	1.03 (1.00–1.06)	0.054	1.03 (1.00–1.06)	0.065
BMI (per kg m^−2^)	0.95 (0.88–1.02)	0.206	0.93 (0.87–1.00)	0.053
Diabetes (yes vs. no)	1.63 (0.84–3.23)	0.154	1.64 (0.86–3.17)	0.128
CVC present (yes vs. no)	0.70 (0.32–1.50)	0.367	0.56 (0.27–1.14)	0.128
HD vintage (per month)	1.04 (0.95–1.13)	0.306	1.05 (0.98–1.11)	0.225
Infection risk (present vs. absent)	1.73 (0.90–3.27)	0.097	1.33 (0.70–2.48)	0.376
Sex (male vs. female)	1.13 (0.59–2.29)	0.722	1.01 (0.53–1.96)	0.982

Note: OR = odds ratio; IRR = incidence rate ratio; CI = confidence interval; BMI = body mass index; CVC = central venous catheter; HD = haemodialysis.

## Data Availability

The de-identified dataset underlying this study exists and is held on secure institutional servers at ULS Coimbra. The raw data supporting the conclusions of this article will be made available by the authors upon reasonable request, subject to institutional approval and data-sharing agreements to protect participant confidentiality.

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
