# Peer review of "Clinical Implementation of an Adapted Infection Risk Screening Tool Following Nurse-Led Haemodialysis Vascular Access Consultation"

_healthcare, 2025, doi:10.3390/healthcare13233058_

Round 1

Reviewer 1 Report

Comments and Suggestions for Authors

The paper is interesting, clear and well written and addresses an import ant subject for the car of dialysis patients. However I see some important questions that should be addressed by the authors.

In the introduction section, I suggest to describe shortly the rope-ladder technique and the buttonhole technique to help readers unfamiliar with hemodialysis to better understand the research question

“An infection risk screening tool is recommended to support this individualised decision”: I understand it is not a recommendation coming from official guidelines. I suggest to rephrase for example by “is recommended by some authors” to avoid ambiguity

To improve readability, I suggest to first describe what is the “nurse-led VA consultation”, what is its content and objective, and at what time of the hemodialysis care pathway it takes place, and then only to mention the performance of the British Infection Risk screening tool

The authors should clarify what type of infection is associated with the buttonhole technique (I suppose bacteremia and local exit AVF site infection?) and what type of infection is assessed by the British Infection Risk screening toolPresently it is only mentioned in the Discussion (“the tool was developed to appraise risk relevant to VA infections”): I suggest it would be useful to mention this important information in the Introduction section. If the BIRST does not take into account other types of infection not directly related to the AVF (such as pneumonia) this would be an important limitation of the possible performance of the BIRST to inform the choice of the puncture technique. In relation with this, I think the implications of a BIRST score “risk present” other than the choice of puncture technique should be more largely explain: presently the authors only briefly mention “the prioritisation of targeted decolonisation and hygiene interventions, and the refinement of risk-stratified care plans in HD units.” ,  More detailed commentd on these topics would help readers understand why the authors studied various types of infection and not only AVF-associated infections

I suggest to develop the possible result and interpretation of the score: the authors suggest it is a binary classification (risk present/absent) but the word “score” suggests an intermediate quantitative result. This should be clarified, especially because the comparatively low performance of the score to predict infection may be explainable by the value of the score used for the binary categorization in the survey

Methods

The fact that during the survey the BIRST was prospectively calculated and its result known to the professional is a very important limitation of the survey, because the fact that actions (for example change of puncture technique or other measures) could therefore be implemented for patients with “risk present” may have reduced infectious risk and therefore falsely decrease the association between “risk present” and the ulterior occurrence of an infection. I think a better design would have been to retrospectively calculate the BIRST in a cohort of patients and to assess if patients with “risk-present” had more infections that patients with “risk absent”. This is in my opinion a major flaw for this study and requires explanations from the authors.

The authors should explain why they should as outcome only infections associated with an hospitalization. And, as mentioned before, it is necessary to explain why all types of infection were considered and not only those directly related with vascular access. This is of importance because of the underlying objective of the survey to assess whether the BIRST could inform the choice of the puncture technique.

I think a previous history of infection since the beginning of hemodialysis was among the variables recorded. Would that not be pertinent?

Discussion:

The fact that the result of BIRST was known to caregivers and probably lead to infection control intervention is a very important limitation of the survey which should be addressed in the Discussion

Author Response

Dear reviewers,

Thank you for your interest in our article entitled “Clinical Implementation of an Adapted Infection Risk Screening Tool Following Nurse-Led Haemodialysis Vascular Access Consultation”, reference code healthcare-3945162.

We would like to thank the reviewers for their constructive comments, which have resulted in a significant improvement to our paper.

Please consider the following table, which contains the reviewers' and editors' comments and our response, point by point. We hope that the paper will now be accepted for publication.

Please do not hesitate to contact us if further information is required.

Best regards

Rui Pinto

===========================================================================

Reviewers' and editors' comments

Authors' response

The paper is interesting, clear and well written and addresses an import ant subject for the car of dialysis patients. However I see some important questions that should be addressed by the authors.

In the introduction section, I suggest to describe shortly the rope-ladder technique and the buttonhole technique to help readers unfamiliar with hemodialysis to better understand the research question

We added a concise paragraph defining rope-ladder (RL), area puncture (AP), and buttonhole (BT), with consistent terminology and supporting references.

“An infection risk screening tool is recommended to support this individualised decision”: I understand it is not a recommendation coming from official guidelines. I suggest to rephrase for example by “is recommended by some authors” to avoid ambiguity

We have rephrased it in accordance with the suggestion:: “An infection risk screening tool is recommended by the British Renal Society to support this individualised decision”

To improve readability, I suggest to first describe what is the “nurse-led VA consultation”, what is its content and objective, and at what time of the hemodialysis care pathway it takes place, and then only to mention the performance of the British Infection Risk screening tool

We added a concise paragraph describing our nurse-led VA consultation—its timing, purpose and core components—thereby clarifying the context in which the British Renal Society Infection Risk Screening Tool is applied

The authors should clarify what type of infection is associated with the buttonhole technique (I suppose bacteremia and local exit AVF site infection?) and what type of infection is assessed by the British Infection Risk screening toolPresently it is only mentioned in the Discussion (“the tool was developed to appraise risk relevant to VA infections”): I suggest it would be useful to mention this important information in the Introduction section.

We have now added a concise paragraph at the end of the Introduction clarifying that infections associated with the buttonhole technique are predominantly AVF site infections and S. aureus bacteraemia, and specifying that the BRS Infection Risk Screening Tool appraises VA-focused infection risk factors to inform BH selection or de-selection.

If the BIRST does not take into account other types of infection not directly related to the AVF (such as pneumonia) this would be an important limitation of the possible performance of the BIRST to inform the choice of the puncture technique. In relation with this, I think the implications of a BIRST score “risk present” other than the choice of puncture technique should be more largely explain: presently the authors only briefly mention “the prioritisation of targeted decolonisation and hygiene interventions, and the refinement of risk-stratified care plans in HD units.” ,  More detailed commentd on these topics would help readers understand why the authors studied various types of infection and not only AVF-associated infections

I suggest to develop the possible result and interpretation of the score: the authors suggest it is a binary classification (risk present/absent) but the word “score” suggests an intermediate quantitative result. This should be clarified, especially because the comparatively low performance of the score to predict infection may be explainable by the value of the score used for the binary categorization in the survey

We have replaced “score” with “screening” throughout and now define the BRS Infection Risk Screening Tool explicitly as a binary checklist (risk present/absent), consistent with its intended clinical use.

Methods

The fact that during the survey the BIRST was prospectively calculated and its result known to the professional is a very important limitation of the survey, because the fact that actions (for example change of puncture technique or other measures) could therefore be implemented for patients with “risk present” may have reduced infectious risk and therefore falsely decrease the association between “risk present” and the ulterior occurrence of an infection. I think a better design would have been to retrospectively calculate the BIRST in a cohort of patients and to assess if patients with “risk-present” had more infections that patients with “risk absent”. This is in my opinion a major flaw for this study and requires explanations from the authors.

We have clarified that screening was conducted once during the hospital-based consultation, that subsequent treatments occurred in regional ambulatory centres, and that the index hospital serves as the regional referral centre for infection-related admissions, ensuring consistent capture of infection-related hospitalisations while acknowledging potential under-ascertainment of outpatient-only events.

The authors should explain why they should as outcome only infections associated with an hospitalization. And, as mentioned before, it is necessary to explain why all types of infection were considered and not only those directly related with vascular access. This is of importance because of the underlying objective of the survey to assess whether the BIRST could inform the choice of the puncture technique.

I think a previous history of infection since the beginning of hemodialysis was among the variables recorded. Would that not be pertinent?

Discussion:

The fact that the result of BIRST was known to caregivers and probably lead to infection control intervention is a very important limitation of the survey which should be addressed in the Discussion

Reviewer 2 Report

Comments and Suggestions for Authors

Suggestions for Enhancements

- It is recommended to elaborate on the rationale behind the selection of the BRS Infection Risk Screening Tool in the Introduction section, particularly in comparison to other available tools.

- Consider incorporating a separate sub-section dedicated to the aims and objectives, it should positioned after the Introduction section and prior to the Materials and Methods section.

- A clarification is required regarding the choice between the Student’s t-test and the Mann–Whitney U test, specifically that the two tests are for parametric and non-parametric tests, respectively. Additionally, include the results of the normality tests.

- Similarly, for the χ2 test or Fisher’s exact test, it is essential to specify which test was applied to which variable for clarity to the readers.

- A separate sub-section for Ethical Considerations should be included, emphasizing adherence to the ethical principles outlined in the Declaration of Helsinki, and this content should be removed from the Study Design and Setting sub-section.

- The STROBE checklist should be submitted as supplementary material for review.

- More comprehensive details regarding the data collection process and the strategies employed for bias management and ensuring data quality should be included.

- In Table 2, clarification is needed regarding the meaning of the codes listed after each cause of infection.

- For Table 4, it is advisable to add a footnote explaining the abbreviated terms used.

- Ensure that the referencing format aligns with the guidelines of the journal, Healthcare.

- Finally, it is important to note that 50% (8 out of 16) of the references are outdated and require updating.

Author Response

Dear reviewers,

Thank you for your interest in our article entitled “Clinical Implementation of an Adapted Infection Risk Screening Tool Following Nurse-Led Haemodialysis Vascular Access Consultation”, reference code healthcare-3945162.

We would like to thank the reviewers for their constructive comments, which have resulted in a significant improvement to our paper.

Please consider the following table, which contains the reviewers' and editors' comments and our response, point by point. We hope that the paper will now be accepted for publication.

Please do not hesitate to contact us if further information is required.

Best regards

Rui Pinto

=================================================================================

Reviewers' and editors' comments

Authors' response

Suggestions for Enhancements

It is recommended to elaborate on the rationale behind the selection of the BRS Infection Risk Screening Tool in the Introduction section, particularly in comparison to other available tools.

We expanded the Introduction to justify our choice of the BRS Infection Risk Screening Tool over generic HAI tools, emphasising its haemodialysis-specific, VA-focused design and its explicit linkage to buttonhole de-selection and prevention actions within a nurse-led consultation pathway.

We conducted an extensive review of the literature and relevant clinical guidelines and found that, to date, no other structured instrument exists that was explicitly designed to assess infection risk for the purpose of guiding cannulation technique selection. Existing tools are either derivative of the BRS checklist or are limited to narrative contraindications. We have clarified this rationale in the Introduction.

Consider incorporating a separate sub-section dedicated to the aims and objectives, it should positioned after the Introduction section and prior to the Materials and Methods section.

To maintain consistency with the typical manuscript structure in Healthcare, we opted not to create a separate subsection. However, to address the reviewer's valid point about clarity, we have consolidated and rewritten the final paragraph of the Introduction to state the study's aims, primary objective, and hypothesis explicitly. We believe this revised paragraph makes the study's purpose more prominent and clearer for the reader.

A clarification is required regarding the choice between the Student’s t-test and the Mann–Whitney U test, specifically that the two tests are for parametric and non-parametric tests, respectively. Additionally, include the results of the normality tests.

We thank the reviewer for this clarification. We now state explicitly in the Statistical Analysis section that the choice between the Student’s t-test and the Mann–Whitney U test depended on the normality of each variable, as assessed by Shapiro–Wilk and visual inspection. We also added the results of the normality tests and specified the test used for each comparison in the table footnotes (Table 1).

Similarly, for the χ2 test or Fisher’s exact test, it is essential to specify which test was applied to which variable for clarity to the readers.

A separate sub-section for Ethical Considerations should be included, emphasizing adherence to the ethical principles outlined in the Declaration of Helsinki, and this content should be removed from the Study Design and Setting sub-section.

A dedicated subsection entitled “Ethical Considerations” has been added under Materials and Methods. It clarifies that the study was conducted in accordance with the Declaration of Helsinki and approved by the Ethics Committee of the Unidade Local de Saúde de Coimbra (PI 2024-ESI-SF.201). The informed consent waiver is also retained in the mandatory “Informed Consent Statement” section, in line with the journal’s submission format.

The STROBE checklist should be submitted as supplementary material for review.

As suggested, we have now submitted the complete STROBE checklist.

More comprehensive details regarding the data collection process and the strategies employed for bias management and ensuring data quality should be included.

In the data collection section, we now describe the data extraction process, cross-validation procedures, and strategies used to minimise selection, classification and measurement bias, as well as the quality checks applied to ensure data integrity.

In Table 2, clarification is needed regarding the meaning of the codes listed after each cause of infection.

We clarified in a table footnote that the codes listed correspond to the ICD-10 primary discharge diagnosis used to classify the cause of infection. As the codes were already listed next to each category, we opted for a concise explanatory note rather than repeating each definition.

For Table 4, it is advisable to add a footnote explaining the abbreviated terms used.

We have added a footnote to Table 4 defining all abbreviated statistical and clinical terms used (OR, IRR, CI, BMI, CVC, HD) to ensure clarity for readers.

Ensure that the referencing format aligns with the guidelines of the journal, Healthcare.

We have reviewed and updated the bibliography to ensure full compliance with the Healthcare journal’s referencing style, including use of the MDPI citation format, sentence case in titles, correct author listings, and consistent formatting of journal names and volume numbers.

Finally, it is important to note that 50% (8 out of 16) of the references are outdated and require updating.

While several of the cited references remain fundamental, we have reviewed the bibliography and updated the manuscript with more recent evidence where applicable.

Reviewer 3 Report

Comments and Suggestions for Authors

The first two keywords do not belong to DECS or MESH terms.
Introduction
It would be appropriate to provide in-depth information on the prevalence of hemodialysis (HD) patients in the country and the infections suffered by this type of patient, in order to obtain a clear picture of the sociodemographic characteristics of the problem being investigated by the authors

Author Response

Dear reviewers,

Thank you for your interest in our article entitled “Clinical Implementation of an Adapted Infection Risk Screening Tool Following Nurse-Led Haemodialysis Vascular Access Consultation”, reference code healthcare-3945162.

We would like to thank the reviewers for their constructive comments, which have resulted in a significant improvement to our paper.

Please consider the following table, which contains the reviewers' and editors' comments and our response, point by point. We hope that the paper will now be accepted for publication.

Please do not hesitate to contact us if further information is required.

Best regards

Rui Pinto

=================================================================================

Reviewers' and editors' comments

Authors' response

The first two keywords do not belong to DECS or MESH terms.

We revised the keywords to MeSH/DeCS-compliant terms.

Introduction
It would be appropriate to provide in-depth information on the prevalence of hemodialysis (HD) patients in the country and the infections suffered by this type of patient, in order to obtain a clear picture of the sociodemographic characteristics of the problem being investigated by the authors

The Introduction section has been expanded to include updated national prevalence data on haemodialysis and vascular access in Portugal, as well as European findings.

Round 2

Reviewer 1 Report

Comments and Suggestions for Authors

I find this revised version clarifies usefully several points of the paper. I think the approach of this work is quite interesting although I am still not fully convinced by the design of the survey (see my first report). However, the authors have added a comment in the Discussion section that mentions the probable attenuation of association due to the design of the survey, which partly takes into account my concern.